# Mitophagy in Acute Kidney Injury and Kidney Repair

**DOI:** 10.3390/cells9020338

**Published:** 2020-02-01

**Authors:** Ying Wang, Juan Cai, Chengyuan Tang, Zheng Dong

**Affiliations:** 1Department of Nephrology, The Second Xiangya Hospital of Central South University, Hunan Key Laboratory of Kidney Disease and Blood Purification, Changsha 410011, China; 168211036@csu.edu.cn (Y.W.); cjane218@csu.edu.cn (J.C.); 2Department of Cellular Biology and Anatomy, Medical College of Georgia at Augusta University and Charlie Norwood VA Medical Center, Augusta, GA 30912, USA

**Keywords:** mitochondria, mitophagy, acute kidney injury, kidney repair

## Abstract

Acute kidney injury (AKI) is a major kidney disease characterized by rapid decline of renal function. Besides its acute consequence of high mortality, AKI has recently been recognized as an independent risk factor for chronic kidney disease (CKD). Maladaptive or incomplete repair of renal tubules after severe or episodic AKI leads to renal fibrosis and, eventually, CKD. Recent studies highlight a key role of mitochondrial pathology in AKI development and abnormal kidney repair after AKI. As such, timely elimination of damaged mitochondria in renal tubular cells represents an important quality control mechanism for cell homeostasis and survival during kidney injury and repair. Mitophagy is a selective form of autophagy that selectively removes redundant or damaged mitochondria. Here, we summarize our recent understanding on the molecular mechanisms of mitophagy, discuss the role of mitophagy in AKI development and kidney repair after AKI, and present future research directions and therapeutic potential.

## 1. Introduction

Acute kidney injury (AKI) is a major kidney disease characterized by rapid decline of kidney function. The main clinical causes of AKI include renal ischemia–reperfusion (IR), sepsis, and nephrotoxins [1,2]. Recent studies have further suggested that preexisting kidney diseases, such as chronic kidney disease (CKD) and diabetic kidney disease (DKD), increase the susceptibility of AKI [3,4]. Lethal and sublethal injury of renal tubules are the main pathological feature of AKI [5]. After AKI, remaining renal tubular cells undergo dedifferentiation, proliferation, immigration, and redifferentiation into mature tubular cells to repair the injured renal tubules. Kidney injury can be completely repaired following mild injury. However, severe or repeated AKI usually results in abnormal repair, leading to renal fibrosis and, eventually, CKD [6,7,8].

Mitochondria are essential organelles for the homeostasis, functions, and viability of eukaryotic cells. In addition to the well-known function of producing adenosine triphosphate (ATP) via oxidative phosphorylation, mitochondria play important roles in maintaining intracellular redox and calcium homeostasis [9]. Moreover, mitochondria serve as a central hub of various signaling pathways for cell survival and death. Thus, mitochondrial damage or dysfunction has been associated with a variety of human diseases, such as neurodegenerative diseases, metabolic diseases, and ischemia–reperfusion injury (IRI) in different organs [10,11,12]. Mitochondrial damage impairs energy production. Damaged mitochondria increase reactive oxygen species (ROS) production, and excessive ROS causes further damage to mitochondria [13], which may stimulate the release of pro-cell death factors, such as cytochrome c (cyt c), from mitochondria into cytosol to initiate cell death pathway [14]. Eukaryotic cells have evolved mitochondrial quality control mechanisms that act at both the molecular and organelle levels to preserve mitochondrial integrity, including mitochondrial protein quality control, mitochondrial antioxidant defense, mitochondrial fission and fusion, mitophagy, and mitochondrial biogenesis (Figure 1) [15,16,17,18]. Mitophagy is a selective form of autophagy that eliminates redundant or damaged mitochondria [19]. Recent evidence suggests that mitophagy plays an important role in AKI development and subsequent kidney repair. In this review, we summarize our recent understanding of the molecular mechanisms of mitophagy, discuss the role of mitophagy in AKI development and kidney repair after AKI, present the therapeutic potential of targeting mitophagy, and propose future research directions in this field.

## 2. Mitochondrial Pathology in AKI Development and Abnormal Repair

The kidney has the second highest mitochondrial content and oxygen consumption in the human body, which is second only to the heart [20]. Therefore, maintaining mitochondrial homeostasis is of crucial importance to normal renal functions. Recent studies have provided substantial evidence that mitochondrial damage and dysfunction contribute critically to AKI development and abnormal kidney repair. Mitochondrial morphological changes including mitochondrial fragmentation, swelling, and loss of inner cristae were observed in renal tubular cells in various experimental models of AKI that was induced by renal IR, sepsis, and nephrotoxins [21,22,23,24]. Consistently, mitochondrial dysfunction in renal tubules, as indicated by mitochondrial oxidative stress and decreased cellular ATP production, was detected in experimental models of AKI [22,25,26,27]. Moreover, mitochondrial damage represents an early event in AKI and occurs before detectable kidney injury. Funk et al. showed that mitochondrial dysfunction occurred before the increase of serum creatinine, a traditional marker of AKI [28]. Brooks et al. further revealed that mitochondrial fragmentation (Figure 2A,B) appeared before renal tubule cell apoptosis during ATP depletion-induced injury in cultured renal tubular cells and in a mouse model of ischemic AKI [29]. Moreover, pharmacological inhibition of mitochondrial fragmentation by mdivi-1, a chemical inhibitor of the mitochondrial fission factor dynamin-related protein 1 (DRP1), before the onset of ischemia dramatically attenuated kidney structural damage and dysfunction [29].

Disruption of mitochondrial biogenesis was also noticed in renal tubular cells in murine models of AKI. Tran M et al. demonstrated the reduction of peroxisome proliferator-activated receptor gamma coactivator 1-alpha (PGC-1α), a major transcription factor for mitochondrial biogenesis, and its downstream genes in renal tubular cells during septic AKI, and importantly ablation of PGC-1α from renal tubule cells suppressed the renal recovery from AKI [30,31]. Collectively, these findings support that mitochondrial pathology contributes critically to the development of AKI.

Recent studies further suggested an involvement of mitochondrial dysfunction in abnormal kidney repair after AKI. Funk et al. showed persistent upregulation of DRP1 and downregulation of the mitochondrial fusion protein mitofusin 2 (MFN2) in renal tubules after renal function recovery in a mouse model of renal IRI [28]. Perry et al. further showed that renal proximal tubule specific deletion of *Drp1* accelerated renal function recovery following renal IR, and moreover, induced deletion of *Drp1* in proximal tubular cells after ischemic AKI dramatically reduced renal fibrosis [32]. These findings suggest that inhibition of DRP1-mediated mitochondrial fragmentation and damage may improve kidney repair after AKI. Moreover, Szeto et al. demonstrated that administration of SS-31, a mitochondrial protective agent, after renal IRI reduced renal fibrosis [33]. In addition, enhancing mitochondrial biogenesis has been demonstrated to accelerate kidney recovery after AKI [34]. Collectively, these findings suggest that mitochondrial pathology is a major mechanism of abnormal kidney repair after AKI.

Mitochondria are a major intracellular source of ROS, and mitochondrial damage increases mitochondrial ROS (mtROS) production. Recent evidence suggests that excessive mtROS production contributes critically to AKI development and abnormal kidney repair [35,36,37]. In line with this notion, a list of mitochondria-targeted antioxidants have been demonstrated to attenuate AKI and accelerate kidney repair, including plastoquinol decylrhodamine 19 (SkQR1), Mito-Tempo, mitoquinolmesylate (MitoQ), and SS-31 [38,39,40,41]. Excessive mtROS cause oxidative damage to mitochondrial components resulting in more ROS production, ultimately increasing the tendency of renal tubular cell death. In contrast to the acute and destructive effect of high levels of mtROS, a moderate increase of mtROS may activate signaling pathways that are involved in AKI pathogenesis and renal fibrosis. For instance, mtROS have been demonstrated to regulate p53, NF-κB, and mitogen-activated protein kinase (MAPK) signaling [42,43,44].

## 3. Overviews of Autophagy

Autophagy is highly conserved lysosomal degradation pathway that removes cytoplasmic components including protein aggregates and organelles [45,46]. Autophagy can be categorized as macroautophagy, microautophagy, and chaperone-mediated autophagy based on the type of cargoes and the route whereby cargoes are delivered to lysosomes [45,46,47]. Macroauphagy is the best characterized form of autophagy and the focus of this review. Macroauphagy (hereafter called autophagy) involves the formation of double-membrane-bound autophagosomes which enclose parts of cytoplasm, and autophagosome-lysosome fusion to form an autolysosome in which cargoes are degraded by lysosomal hydrolase, and the degradation products are released for recycling [47,48]. Functionally, basal autophagy under physiological conditions is essential for maintaining cellular homeostasis. Under pathological conditions, autophagy acts as an adaptive or defense mechanism to preserve cell viability.

Autophagy is a tightly regulated process. The core autophagy machinery that is constituted of 6 protein complexes of autophagy-related proteins (Atg) regulate autophagy at different levels [49,50]. The Atg1/unc-51 like autophagy activating kinase 1 (ULK1) protein complex is the key regulator of autophagy initiation [49]. The class III phosphatidylinositol 3-kinase (PtdIns3K) complex comprising phosphatidylinositol 3-kinase catalytic subunit type 3 (PIK3C3/VPS34), PIK3R4/VPS15, Beclin-1 (BECN1) and ATG14L regulates autophagy initiation and autophagosome maturation via interacting with several regulatory proteins [51]. PtdIns3P-binding proteins WD repeat domain phosphoinositide-interacting proteins and double FYVE-containing protein 1 and ATG9L regulate membrane transfer from surrounding sources to the expanding phagophore [52]. Two ubiquitin-like conjugation systems, the ATG12–ATG5-ATG16L complex and the microtubule-associated protein 1 light chain 3–phosphatidyl ethanolamine(MAP1LC3/LC3–PE), regulate the extension and completion of autophagosome [53]. Autophagy is tightly regulated to enable the cell to maintain an optimal balance between synthesis and degradation, use and recycling of cellular components. Recent evidence suggests the nutrient/energy pathways including mechanistic target of rapamycin complex 1 (MTORC1), protein kinase AMP-activated catalytic subunit alpha 1 (AMPK), and sirtuin 1 (SIRT1) are major upstream regulators of autophagy [54]. In addition, a variety of cellular stress including oxidative stress, ER stress, hypoxia, DNA damage, and immune signals have also been implicated as important regulators of autophagy [54,55].

## 4. Mitophagy

Autophagy has long been thought of as a non-selective bulk degradation pathway [56,57], but recent studies have demonstrated that autophagy can selectively eliminate specific cargoes, such as proteins aggregates, endoplasmic reticulum (ER), lipids, and mitochondria [58,59,60]. Mitophagy is a selective form of autophagy that specifically eliminates superfluous or damaged mitochondria. Under physiological conditions, mitophagy has been shown to remove superfluous mitochondria during erythrocyte maturation, and remove sperm-derived mitochondria during the development of fertilized oocytes [61]. Under stressful conditions, mitophagy is induced as an adaptive or defense mechanism for maintaining a population of healthy mitochondria and thereby cell survival. So far, defects in mitophagy have been associated with a variety of human diseases, such as neurodegenerative diseases, metabolic diseases, ischemia–reperfusion injury and so on [25,62,63].

Mitophagy requires efficient recognition of targeted mitochondria and subsequent enclosure of mitochondria within autophagosomes. Currently, two major mechanisms of mitophagy have been proposed, including ubiquitin (Ub)-dependent and -independent mechanisms. The Ub-dependent mechanism is regulated by PTEN-induced putative kinase 1 (PINK1)-parkin RBR E3 ubiquitin protein ligase (PARK2) pathway. The Ub-independent mechanism is regulated by mitophagy receptors that localize on mitochondrial outer membrane, including BCL2 interacting protein 3 (BNIP3), BNIP3-like (BNIP3L/NIX), and FUN14 domain containing 1 (FUNDC1) (Figure 3) [64,65,66].

### 4.1. PINK1-PARK2 Pathway of Mitophagy

PINK1-PARK2 pathway is the most well-documented mechanism for mitochondrial identification and labelling during mitophagy. PINK1 and PARK2 are initially identified as Parkinson’s disease-related proteins. PINK1 is a mitochondrial serine/threonine kinase [67], and PARK2 is a cytosolic ubiquitin E3 ligase [68]. Under healthy conditions, PINK1 is constitutively imported into mitochondria, where it is cleaved by the intramembrane serine protease presenilin associated rhomboid like (PARL) and ultimately degraded [69]. Under stressful conditions, the loss of mitochondrial membrane potential impedes the import of PINK1 into mitochondria, resulting in PINK1 accumulation at mitochondrial outer membrane (MOM). PINK1 on MOM recruits PARK2 from cytosol to damaged mitochondria and activates PARK2 E3 ligase activity through directly phosphorylating PARK2 and ubiquitin. Active PARK2 builds ubiquitin chain on MOM proteins. The ubiquitin-labeled mitochondria are subsequently recognized by autophagy receptor proteins, such as voltage-dependent anion channel 1 (VDAC1) and sequestosome1 (p62/SQSTM1) that link mitochondria to autophagosomes through interacting with LC3 in autophagosome membrane, leading to autophagic engulfment of mitochondria for degradation [70,71]. In addition, Lazarou et al. demonstrated that PINK1-mediated phosphorylation of Ub was sufficient to recruit autophagy receptors NDP52 and optineurin (OPTN) to induce mitophagy regardless of PARK2 [72].

### 4.2. BNIP3 and NIX Mediated Mitophagy

BNIP3 and NIX are BH3-only proteins that localize in MOM [73]. They were initially identified as pro-apoptotic proteins [74,75]. Recent studies indicate that they are also mitophagy regulators. Under the condition of hypoxia, BNIP3 and NIX are transcriptionally activated by HIF hypoxia-inducible factor-1 (HIF-1) and/or forkhead homeobox type O (FOXO) to induce autophagic removal of damaged mitochondria [76]. Mechanistically, BNIP3 and NIX act as mitophagy receptors that bridge mitochondria to autophagosome by directly interacting with LC3 in autophagosome membrane via their LC3-interacting regions (LIRs) [77]. Emerging evidence suggests that BNIP3 can also induce mitochondrial fragmentation that is pre-requisite for mitophagy, and competitively bind with BCL2 proteins to release BECN1 to induce autophagy [78]. Besides functioning as a mitophagy receptor, NIX was shown to interact with small GTPase to activate mitophagy [79].

### 4.3. FUNDC1-Mediated Mitophagy

FUNDC1 localizes in MOM, and can also link mitochondria to autophagosomes through its direct interaction with LC3 via its LIR domains. Phosphorylation plays an important role in regulating the pro-mitophagy function of FUNDC1. Under normal circumstances, the protein kinase SRC (SRC proto-oncogene, non-receptor tyrosine kinase) phosphorylates FUNDC1 to disrupt its interaction with LC3. Upon hypoxia, SRC activity is inhibited, which in turn reduces FUNDC1 phosphorylation and thereby increase FUNDC1 interaction with LC3 to induce mitophagy [80,81,82]. Overexpression of FUNDC1 was also shown to stimulate mitophagy [83]. Notably, FUNDC1 can also regulate mitochondrial dynamics via interacting with DRP1 or OPA1 (OPA1 mitochondrial dynamin-like GTPase), respectively. FUNDC1 phosphorylation at Ser13 facilitates its interaction with OPA1 to inhibit mitochondrial fission, whereas FUNDC1 dephosphorylation at Ser13 releases FUNDC1 from OPA1 and promotes its interaction with DRP1 to induce mitochondrial fission and mitophagy under cell stress [84,85].

## 5. Mitochondrial Fission and Fusion

Mitochondria are dynamic organelles undergoing constant fission and fusion. Recent studies suggest that fission is required for mitophagy [86]. Fission may yield two types of mitochondria, i.e., polarized and depolarized daughter mitochondria [87]. The depolarized daughter mitochondria have decreased expression of OPA1 and thus have a reduced probability to fuse, and will be removed by mitophagy. Under stressful conditions, fission isolates damaged parts from mitochondrial network for autophagic degradation [88,89]. As such, fission facilitates mitophagy, whereas suppressing fission or heightening fusion inhibits mitophagy. Mitochondrial fission is regulated by DRP1 and its receptor proteins including mitochondrial fission factor (Mff), fission-1 (Fis1), and mitochondrial kinetic proteins 49 and 51 (Mid49/51) [90,91]. Under stressful conditions, DRP1 is translocated from the cytosol to mitochondria via interacting with its receptor proteins, where DRP1 oligomerizes and contracts around the mitochondria to cleave the organelles. Multiple post-translational modifications, including phosphorylation, dephosphorylation, s-nitrosylation, SUMOylation, and ubiquitination, play important roles in regulating DRP1 activity. In the kidney, knockdown of *Drp1* in cultured renal tubular cells inhibited mitochondrial division and mitophagy [92]. Consistently, pharmacological inhibition of fission suppressed renal IR-induced mitophagy in renal tubule cells in a mouse model of renal IRI [93]. These findings support that mitochondrial fission is required for mitophagy in AKI.

Mitochondrial fusion involves the fusions of outer mitochondrial membranes and inner mitochondrial membranes that are mainly regulated by mitofusin 1 and 2 (MFN1, MFN2), and OPA1, respectively. Emerging evidence suggests that MFN1 and MFN2 are substrates of PARK2 E3 ligase during PINK1-PARK2 pathway of mitophagy [94]. Recent studies also demonstrated MFN2 as a connecting molecule between mitochondria and ER at ER-mitochondria contacts, a structure may promote the formation of mitochondrial autophagosomes and mitophagy [95,96,97,98].

## 6. Mitochondrial Biogenesis

Mitochondrial biogenesis (MB) refers to the generation of new mitochondrial mass and replication of mitochondrial DNA through the proliferation of pre-existing organelles. MB is essential for meeting the increase of cellular energy requirement and repopulating mitochondrial contents in newly generated cells during cell proliferation [99]. It is also critical for replacing damaged and dysfunction mitochondria that are selectively removed by mitophagy [100]. Mitochondrial proteome is composed by proteins encoded by both mitochondrial DNA and nuclear DNA [101]. Thus, coordinated expression of mitochondrial proteins encoded by mitochondrial and nuclear DNA is critically important for mitochondrial biogenesis. Mitochondrial biogenesis is mainly regulated by PGC-1α (peroxisome proliferator-activated receptor γ coactivator-1α), which directly regulates an array of transcription factors to modulate expression of nuclear genes that are involved in MB, including nuclear respiratory factor 1 (NRF-1), NRF-2, peroxisome proliferator-activated receptor alpha, estrogen-related receptor alpha, YY1 transcription factor, among others [102]. AMP-activated protein kinase (AMPK) and NAD-dependent deacetylase sirtuin-1 are major upstream regulator of PGC-1α which activate PGC-1α by phosphorylation and deacetylation modification, respectively [103]. Recent studies have provided substantial evidence that enhancing MB in renal proximal tubular cells facilitates kidney recovery and complete kidney repair after AKI [104].

## 7. Mitophagy in AKI

Defective mitophagy has been implicated in the pathogenesis of a variety of human illnesses including neurodegenerative diseases, metabolic diseases, and cardiovascular diseases [99,105,106]. Similarly, the role and regulation of mitophagy in AKI has attracted lots of attention in recent years. These studies demonstrate that mitophagy activation in renal tubular cells represents a renoprotective mechanism during AKI (Table 1).

### 7.1. Mitophagy in Renal Ischemia–Reperfusion Injury

Renal IR is one of the common causes of AKI that is often related with renal vascular occlusion, kidney transplantation, and cardiac surgery, etc. [117]. Mitophagy is induced in renal tubular cells, particularly in renal proximal tubular cells, following renal IR. Ishihara et al. showed that BNIP3 was upregulated in a HIF1-dependent manner during renal IRI. They further showed that in cultured renal tubular cells, mitophagy was increased by BNIP3 overexpression and suppressed by BNIP3 knockdown, suggesting a possible occurrence of BNIP3-dependent mitophagy in ischemic AKI [107]. Recent studies from others and us further demonstrated that mitophagy plays a protective in ischemic AKI. Li et al. showed that pharmacological inhibition of DRP1 by mdivi-1 dramatically reduced renal IR-induced mitophagy, resulting in the aggravation of renal dysfunction [93]. Our recent study further revealed an increased expression of PINK1 and PARK2, accompanied by mitophagy activation in cultured renal tubular cells following ATP depletion–repletion injury and in renal tubules of mouse models of ischemic AKI (Figure 2C). Knockdown of *Pink1* or *Park2* partially inhibited mitophagy activation in cultured renal tubular cells subjected to ATP depletion–repletion, and enhanced cell apoptosis. Importantly, renal IR-induced mitophagy was partially abrogated in renal tubules of *Pink1* and *Park2* single or double knockout. Moreover, *Pink1* and *Park2* single or double knockout enhanced the accumulation of damaged mitochondria in tubular cells, ROS production, and inflammation, resulting in more severe kidney injury. Together, these findings suggest that PINK1-PARK2 pathway of mitophagy has a protective role in preserving renal tubular integrity and normal renal function in ischemic AKI [108]. More recently, we further demonstrated an involvement of BNIP3-mediated mitophagy in renal tubular cells in ischemic AKI. BNIP3 was induced in cultured renal tubular cells subjected to oxygen-glucose deprivation and reoxygenation and in renal tubules of a mouse model of renal IRI. Functionally, BNIP3 deficiency inhibited mitophagy, accumulated damaged mitochondria, and enhanced tubular cell death, thereby aggravating renal IRI. Collectively, these findings suggest that different pathways of mitophagy are activated as renoprotective mechanisms in ischemic AKI [109]. Our latest studies also demonstrated that mitophagy induction in renal tubular cells may contribute critically to the beneficial effect of ischemic preconditioning (IPC). IPC and autophagy activation by BECN1 peptide enhanced mitolysosome formation during renal IRI in mitophagy reporter mice. *Pink1* knockdown in renal tubular cells abolished IPC-related mitophagy and the protective effect of IPC, suggesting the contribution of PINK1-mediated mitophagy in renal protection afforded by IPC [110]. In addition, Feng et al. revealed that mammalian STE20-Like Kinase 1 (Mst1) was a negative regulator of mitophagy that suppressed mitophagy by inactivating AMPK signaling pathway and downregulating OPA1 expression in renal tubular cells [111]. Mst1 was upregulated in renal tubules following renal IRI, and Mst1 deficiency dramatically attenuated renal IRI. In addition, Wei et al. noticed a close association between mitophagy induction and reduced PPARγ expression in cultured renal tubular cells during hypoxia/reoxygenation (HR), but the potential role of PPARγ in the regulation of mitophagy under this condition remained unknown [112]. Together, these findings suggest that mitophagy acts an important mechanism for preserving mitochondrial quality and tubular cell survival in ischemic AKI.

### 7.2. Mitophagy in Nephrotoxin-Induced AKI

Mitophagy activation and the renoprotective role of mitophagy have also been demonstrated in nephrotoxin-induced AKI, especially in experimental models of cisplatin nephrotoxicity. Cisplatin is a chemotherapy drug with nephrotoxicity or side-effects in kidneys. Recent studies have demonstrated mitophagy activation in renal tubular cell during cisplatin nephrotoxicity. Yuan et al. showed that mitophagy was induced in culture renal tubular cells exposed to cisplatin. Cisplatin-induced mitophagy was partially abrogated by knockdown of *Pink1* or *Park2*, but was enhanced by overexpression of *Pink1* or *Park2*. In addition, suppressing PINK1 or PARK2 expression increased the sensitivity of renal tubular cells to cisplatin-induced cell death [113]. In a subsequent study, they further demonstrated that DRP1-mediated mitochondrial fragmentation was essential for mitophagy in this condition [92]. Our recent study confirmed the activation of PINK1-and PARK2-dependent mitophagy in renal tubules during cisplatin nephrotoxicity. Using *Pink1* or *Park2* deficient mice, we provided evidence that PINK1 and PARK2-dependent mitophagy was protective against kidney injury during cisplatin nephrotoxicity [114]. In contrast, a recent study by Zhou et al. showed that PINK1 deficiency ameliorated cisplatin-induced mitochondrial fragmentation, mitophagy, and kidney injury in rats [115]. The cause of the discrepancy between these studies remains unknown. Emerging evidence also suggests a potential role of BNIP3 in regulating mitophagy in cisplatin-induced AKI. Upregulation of BNIP3 was noticed in renal tubules of rats exposed to cisplatin [115]. Liang et al. showed panax notoginseng saponins enhanced mitophagy via a HIF-1α/BNIP3/BENCI signaling pathway to protect against cisplatin-induced nephrotoxicity [116]. Together, these findings support a protective role of mitophagy in maintaining tubular cell integrity and promoting cell survival in cisplatin-induced AKI.

### 7.3. Mitophagy in Sepsis-Associated AKI

Sepsis is a life-threatening, complex syndrome that results from bacterial invasion and systemic inflammation. It is one of the most common reasons of AKI [118]. Activation of mitophagy has been implicated in septic AKI. Takasu and colleagues showed that the autophagosomes in renal tubular cells in the renal tissue of sepsis patients were juxtaposed with aquatic mitochondria with cristae damage, indicating mitophagy activation in this condition [119]. In a mouse model of septic AKI that was induced by cecal ligation and puncture (CLP), mitophagy in proximal renal tubules was induced at 4 h after CLP, but decreased at 18 to 24 h after CLP, indicating a dynamic change of mitophagy during septic AKI [120]. More recently, Dai et al. revealed that mitophagy was activated in cultured renal tubular cells at 2 h after lipopolysaccharide (LPS) treatment, and in renal tubules of mice at 2 h after CLP [121]. They further demonstrated that PINK1 knockdown dramatically reduced LPS-induced mitophagy, and enhanced LPS-induced cell apoptosis, suggesting that PINK1-mediated mitophagy was protective against septic AKI [121]. Despite these studies, the molecular mechanisms underlying mitophagy activation in sepsis-induced AKI remain largely unknown.

### 7.4. Mitophagy in Other Types of AKI

Mitophagy has also been implicated in AKI that is induced by other risk factors, such as contrast media (CI) and folic acid (FA). Contrast agent is one of the common causes of hospital-acquired AKI [122]. Lei et al. firstly showed that iodinated contrast media-induced mitophagy in cultured renal tubular cells to protect against cell apoptosis [123]. Their subsequent study demonstrated that mitophagy was also activated in renal tubules in a rat model of CI-induced AKI [124]. Consistently, a recent study by Lin et al. showed mitophagy was induced in cultured renal tubular cells exposed to CI and in renal tubules in a mouse model of CI-induced AKI. They provided further evidence that *Pink1 or Park2* deletion suppress CI-induced mitophagy in renal tubules, and loss of PINK1-PARK2 pathway of mitophagy aggravated tubular cells apoptosis and tissue damage accompanied with enhanced mtROS production, DNA oxidative injury, and activation of NLRP3 (NLR family pyrin domain containing 3) inflammasome [35]. In addition, rapamycin upregulated mitophagy and attenuated mitochondrial injury and oxidative stress [125]. These findings suggest a renoprotective role of PINK1-PARK2 pathway of mitophagy in CI-induced AKI. The latest work by Gong et al. verified the activation of mitophagy in CI-induced AKI in rats. The natural anti-oxidant tetramethylpyrazine suppressed ROS production, mitophagy, mitochondrial fragmentation, and renal tubular cell apoptosis in CI-induced AKI [126]. In addition, Mitophagy was also noticed in renal tubules in FA-induced AKI in rats [127], and in diatrizoic acid (DA)-induced renal tubular cell injury [128], but the precise role and regulation of mitophagy in these settings remain unknown.

## 8. Mitophagy in Kidney Repair and Renal Fibrosis

After kidney injury, survival tubular cells undergo dedifferentiation, proliferation, immigration, and redifferentiation into mature tubular cells to restore injured renal tubules [6]. Abnormal kidney repair results in renal fibrosis and, eventually, CKD. As mentioned above, mitochondrial damage contributes critically to delayed or abnormal kidney repair after AKI [129]. Thus, timely removal of injured mitochondria may facilitate kidney repair. In line with this notion, Lan et al. demonstrated a reduction of mitochondrial number accompanied by increased glycolysis in regenerated renal tubules after AKI [130]. Notably, this mitochondrial change was reversed in normal repaired tubules but persisted and became progressively more severe in tubule cells that failed to dedifferentiate, suggesting a potential involvement of mitophagy in kidney repair after AKI. Nonetheless, the regulation and precise function of mitophagy in kidney repair after AKI awaits future investigation. Recent studies also demonstrated an involvement of mitophagy in unilateral ureter obstruction (UUO)-induced renal fibrosis. Su et al. demonstrated that NLRP3 negatively regulated mitophagy in renal tubular cells through an inflammasome-independent manner under the condition of hypoxia. Moreover, UUO induced higher levels of mitophagy in renal tubules of NLRP3 knockout mice than in their wildtype littermates. They provide further evidence that NLRP3 knockout upregulated mitophagy to impede the progression of CKD [37]. More recently, Bhatia et al. report that PINK1-MFN2-PARK2 pathway of mitophagy in macrophages was compromised in experimental and human kidney fibrosis. Failure of this pathway of mitophagy resulted in accumulation of abnormal mitochondria, increased mtROS production, and increased RICTOR (RPTOR independent companion of MTOR complex 2) expression in macrophages, which in turn promoted the differentiation of macrophages toward profibrotic/M2 phenotype, leading to higher extracellular matrix production and progression of kidney fibrosis in UUO mice, suggesting that PINK1-MFN2-PARK2-mediated mitophagy in macrophages plays an anti-fibrotic role in kidneys [131]. Collectively, these findings suggest that mitophagy in renal tubular cells facilitates renal repair.

## 9. Conclusions

Recent studies have provided substantial evidence that mitophagy is activated in renal tubular cells and plays a protective role in AKI. As such, pharmacological augmentation of mitophagy represents a promising therapeutic strategy for the prevention and treatment of AKI. Despite these interesting findings, questions remain open. For instance, what`s the role of mitophagy in different types of kidney cells in AKI? What mechanisms lead to mitophagy activation in renal tubular cells? Mitophagy has been implicated in a variety of cellular processes, such metabolism, inflammation, and cell proliferation, but the underlying mechanism remains unclear. In addition, specific activators of mitophagy are not currently available. In the recovery phase of AKI, the role of mitophagy is largely unclear and little is known about its regulation. A detailed understanding of the regulation and pathological effects of mitophagy in AKI and kidney repair will facilitate the discovery of new therapeutic approaches for the treatment of AKI and prevention of the progression of AKI to CKD.

## Figures and Tables

**Figure 1 cells-09-00338-f001:**
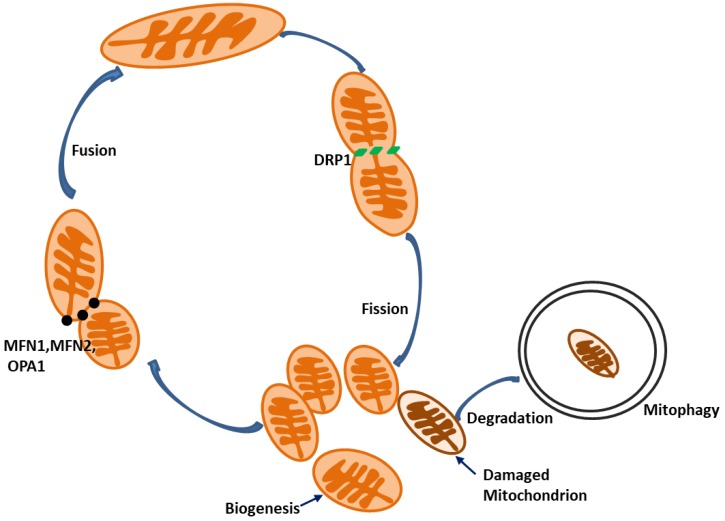
Mitochondrial quality control. Mitochondrial dynamics, mitophagy and biogenesis are important mechanisms of mitochondrial quality control. Mitochondrial dynamics includes two opposing processes: mitochondrial fusion and fission. Mitochondrial fission is regulated by DRP1, whereas fusion is regulated by mitofusin 1 (MFN1), MFN2, and OPA1. Damaged or depolarized mitochondria are isolated from mitochondrial network by fission and are then degraded through mitophagy. Mitochondrial biogenesis increases mitochondrial mass to meet increased energy demand and/or replace the mitochondria that have been removed by mitophagy.

**Figure 2 cells-09-00338-f002:**
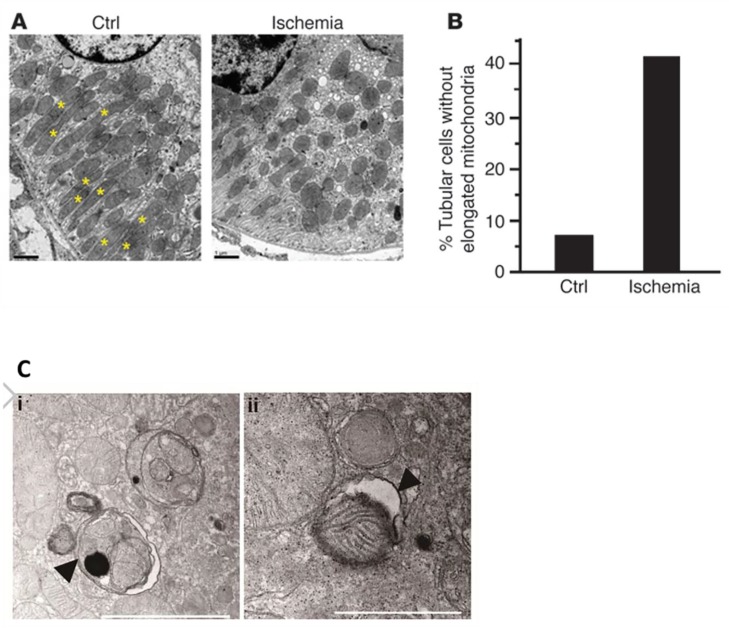
Mitochondrial fragmentation and mitophagy induction in proximal tubule cells during renal ischemia–reperfusion injury. (**A**) Mitochondrial fragmentation was observed in proximal tubule cells immediately after renal ischemia–reperfusion (IR). The fragmentation was remarkably increased in IR kidney. Asterisks indicate elongated (>2 μm) mitochondria [29]. (**B**) Quantification of mitochondrial fragmentation [29]. (**C**) Representative TEM images of autophagosomes (pointed to by the arrowhead in the left panel) and a mitophagosome (pointed to by the arrowhead in the right panel) in renal proximal tubule cells after IR [45].

**Figure 3 cells-09-00338-f003:**
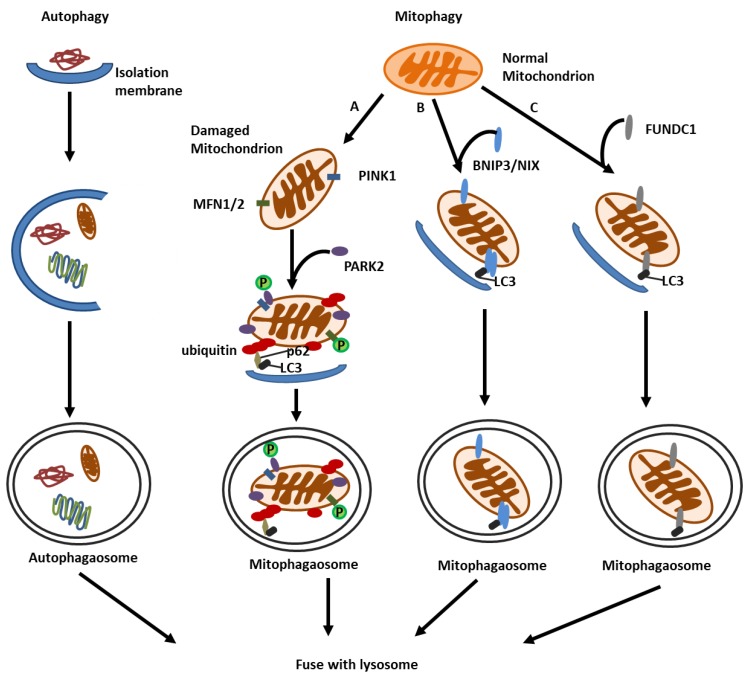
Molecular mechanisms of autophagy and mitophagy. Autophagy involves the formation of isolation membrane, its extension and enclosing of cytoplasmic contents to form the autophagosome, and fusion of the autophagosomes with lysosomes to form the autolysome. Mitophagy is a selective form of autophagy for the clearance of superfluous or damaged mitochondria. There are three well-characterized pathways of mitophagy: (A) PINK1-PARK2 pathway. Under the condition of mitochondrial depolarization, PINK1 accumulates on the mitochondrial outer membrane (MOM), where it phosphorylates and recruits PARK2 E3 ligase to mitochondria and build uibiquitin chains on MOM proteins. The ubiquitinated proteins recruit receptor proteins, such as p62/SQSTM1, which link the ubiquitin-labelled mitochondria to LC3 in autophagosome membrane, leading to the formation of mitophagosome and mitochondrial degradation. BNIP3, NIX(B) and FUNDC1(C) are MOM proteins that directly bridge mitochondria with LC3 in autophagosome membrane to form mitophagosome.

**Table 1 cells-09-00338-t001:** Summary of the studies of mitophagy in acute kidney injury (AKI).

AKI Categories	Pathways that Regulate Mitophagy	Roles on AKI	Mechanisms	References
Renal Ischemia- Reperfusion	P53/sestrin-2 pathway, HIF-1α/BNIP3 pathway	Protect against AKI	Modulate cell apoptosis	[107]
	DRP1-dependent pathway	Protect against AKI	Eliminate damaged mitochondria	[94]
	PINK1/PARK2 pathway	Protect against AKI	1. Eliminate damaged mitochondria	[45,108]
			2. Modulate the removal of ROS	
			3. Relieve inflammatory response	
			4. Suppress mitochondrial depolarization	
			5. Improve ATP production	
	BNIP3-mediated pathway	Protect against AKI	1. Remove damaged mitochondria	[109]
			2. Modulate the elimination of ROS	
			3. Relieve inflammatory response	
	OPA1-related pathway	Protect against AKI	1. Degrade the damaged mitochondria	[110]
			2. Interrupt the mitochondrial damage signalling	
Cisplatin-induced AKI	PINK1/PARK2 pathway	Protect against AKI	1. Suppress apoptosis	[111,112,113]
			2. Protect mitochondrial function	
			3. Inhibit Drp1-mediated mitochondrial fission	
	DRP1-dependent pathway	Protect against AKI	Inhibit mitochondrial dysfunction	[93]
	HIF-1α/BNIP3/BCEN-1 pathway	Protect against AKI	Alleviate apoptosis	[114]
Spesis-induced AKI	PINK1/PARK2 pathway	Protect against AKI	Prevent cell apoptosis	[115]
Contrast media-Induced AKI	PINK1/PARK2 pathway	Protect against AKI	1. Reduce mitochondrial ROS	[35,116]
			2. Inhibit NLRP3 inflammasome activation	

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
