# Peer review of "Mitophagy in Acute Kidney Injury and Kidney Repair"

_cells, 2020, doi:10.3390/cells9020338_

Round 1

Reviewer 1 Report

Wang et al present a very good and up-to-date review about the role of mitophagy and AKI. The article covers all relevant literature, is unbiased, and highlights the importance of further understanding its role so that new treatments for disease can be developed.

The diagrams are very helpful in understanding the topic and are much appreciated. However, with regards to Figure 3, it is not clear if the authors have received permission to reproduce the photomicrographs and bar graphs from Brooks et al, JCI, 2009.

Lastly, while I think the article is worthy of publishing, it would be good if a copy-editor could go through the article and correct a few grammatical errors so that the readership would better appreciate the paper.

Author Response

Reviewer 1:

Wang et al present a very good and up-to-date review about the role of mitophagy and AKI. The article covers all relevant literature, is unbiased, and highlights the importance of further understanding its role so that new treatments for disease can be developed.

The diagrams are very helpful in understanding the topic and are much appreciated. However, with regards to Figure 3, it is not clear if the authors have received permission to reproduce the photomicrographs and bar graphs from Brooks et al, JCI, 2009.

Response: Thank you for the suggestion. We have the permission to reproduce the images in Figure 3.

Lastly, while I think the article is worthy of publishing, it would be good if a copy-editor could go through the article and correct a few grammatical errors so that the readership would better appreciate the paper.

Response: Thanks for the nice suggestion. We have corrected the grammatical errors and have further polished the writing for language use.

Reviewer 2 Report

The review article by Wang et al describes a key role of mitochondrial pathology in AKI development and abnormal kidney repair after AKI. Furthermore, they describe that a timely elimination of damaged mitochondria in renal tubular cells represents an important quality control mechanism for cell homeostasis and survival during kidney injury and repair. In addition, they summarize the recent understanding on the molecular mechanisms of mitophagy, discuss the role of mitophagy in AKI development and kidney repair after AKI, and present future research directions and therapeutic potential.

This is an interesting review about the role of mitophagy in AKI. I have the following comments:

The role of ROS in AKI and specially in mitophagy should be expanded. The development of AKI in some conditions such as type 2 diabetes should be described. The treatment with antioxidants in AKI and molecular pathways associated should be expanded and described.

There are some mistakes along the manuscript such as the numbers in the figure legends.

Author Response

The review article by Wang et al describes a key role of mitochondrial pathology in AKI development and abnormal kidney repair after AKI. Furthermore, they describe that a timely elimination of damaged mitochondria in renal tubular cells represents an important quality control mechanism for cell homeostasis and survival during kidney injury and repair. In addition, they summarize the recent understanding on the molecular mechanisms of mitophagy, discuss the role of mitophagy in AKI development and kidney repair after AKI, and present future research directions and therapeutic potential.

This is an interesting review about the role of mitophagy in AKI. I have the following comments:

The role of ROS in AKI and specially in mitophagy should be expanded. The development of AKI in some conditions such as type 2 diabetes should be described. The treatment with antioxidants in AKI and molecular pathways associated should be expanded and described.

Response:

Thanks for the suggestion. We have added the content on the role and potential action mechanism of mitochondrial ROS in the development of AKI, and also mentioned the antioxidants that have been used in experimental AKI. Please see the last paragraph in the section of “2.Mitochondrial Pathology in AKI Development and Abnormal Repair”

We have added the description of “Recent work has further suggested that preexisting kidney diseases, such as chronic kidney disease (CKD) and diabetic kidney disease (DKD) increase the susceptibility of AKI” in the 1st paragraph in the section of “Introduction”

There are some mistakes along the manuscript such as the numbers in the figure legends.

Response: We have corrected these mistakes in the revision.

Reviewer 3 Report

Wang et al. give a comprehensive overview on the role of mitochondria in general and specifically regarding their role in acute kidney injury.

General issues:

The manuscript suffers from many smaller grammatical errors (singular/plural; missing articles) and should be proof-read by a native speaker or a language editing service.

All abbreviations should be written in full length and explained when mentioned first. This is not done consistently.

Do the authors have histological or EM pictures of their animal models so the morphology of AKI, its repair and changes of the mitochondria could be demonstrated to the readership? This would surely improve the present manuscript significantly.

Specific issues:

Figure 3: The yellow stars are not explained. Two or more EM pictures with higher magnifications would improve the figure markedly.

Line 96-100: is Drp1 now beneficiary or deleterious? This should be explained due to the discrepant study results

Line 151: what should “et al” mean here?

Paragraph 4.1.: many grammatical errors

SUMOlyation should be written like this and also explained to the readership.

Mitochondrial biogenesis: how this works should be explained in a few more sentences.

Line 261: should et al be replaced by etc.?

Line 376: prevent is not correct here, maybe inhibit would be better

Author Response

Wang et al. give a comprehensive overview on the role of mitochondria in general and specifically regarding their role in acute kidney injury.

General issues:

The manuscript suffers from many smaller grammatical errors (singular/plural; missing articles) and should be proof-read by a native speaker or a language editing service.

Response: We have corrected the grammatical errors and have further polished the writing for language use.

All abbreviations should be written in full length and explained when mentioned first. This is not done consistently.

Response: Thank you for the suggestion. We have added full spelling and explanation for all abbreviations at the first mention.

Do the authors have histological or EM pictures of their animal models so the morphology of AKI, its repair and changes of the mitochondria could be demonstrated to the readership? This would surely improve the present manuscript significantly.

Response: We agree, but unfortunately we do not have such pictures at this moment.

Specific issues:

Figure 3: The yellow stars are not explained. Two or more EM pictures with higher magnifications would improve the figure markedly.

Response: We have added an explanation for the yellow stars. These EM images show clear mitochondrial fragmentation (Figure 3A) and the formation of autophagosome and mitophagosome (Figure 3A).They are most representative images that we have for this experimental setting.

Line 96-100: is Drp1 now beneficiary or deleterious? This should be explained due to the discrepant study results

Response: DRP1 and associated mitochondrial fragmentation appear deleterious in this experimental setting. We have changed the description into “Perry et al. further showed that renal proximal tubule specific deletion of Drp1 accelerated renal function recovery following renal IR, and moreover, induced deletion of Drp1 in proximal tubular cells after ischemic AKI dramatically reduced renal fibrosis[30]. These findings suggest that inhibition of DRP1-mediated mitochondrial fragmentation may improve kidney repair after AKI.”

Line 151: what should “et al” mean here?

Response: We have changed “et al” to “etc.” Thanks.

Paragraph 4.1.: many grammatical errors

SUMOlyation should be written like this and also explained to the readership.

Response: We have made the changes as suggested. SUMOylation is a process in which SUMO proteins are covalently attached to specific lysine residues in target proteins.

Mitochondrial biogenesis: how this works should be explained in a few more sentences.

Response: We have changed the description of mitochondrial biogenesis as “Mitochondrial biogenesis (MB) refers to the generation of new mitochondrial mass and replication of mitochondrial DNA through the proliferation of pre-existing organelles”

Line 261: should et al be replaced by etc.?

Response: Thanks. We have made the changes as suggested.

Line 376: prevent is not correct here, maybe inhibit would be better.

Response: We have made the changes as suggested.

Round 2

Reviewer 2 Report

no more comments

Author Response

Thanks, we also feel this review is a good addition to the specific issue of "Selective Autophagy".

Reviewer 3 Report

The authors considerably improved the manuscript yet I am still not very happy with the EM figure. Have the authors formally asked for permission to publish the figure? I would suggest to look for another figure taken from the literature that Shows a higher magnification.

Author Response

"The authors considerably improved the manuscript yet I am still not very happy with the EM figure. Have the authors formally asked for permission to publish the figure? I would suggest to look for another figure taken from the literature that Shows a higher magnification."

We thank the reviewer for recognizing the improvement of the revised manuscript. We agree that the EM images do not have the highest quality. We chose to use these images because they have been formally published and they are sufficient to support the idea presented in this review article. In addition, these EM images were produced and published by our (Dr. Dong's) laboratory. Yes, we have the permissions from both Journal of Clinical Investigation and Autophagy to re-use these images as the original author.